# Adsorption Analysis of Exopolymeric Substances as a Tool for the Materials Selection of Photobioreactors Manufacture

**DOI:** 10.3390/ijms232213924

**Published:** 2022-11-11

**Authors:** Lucía García-Abad, Yolanda Soriano-Jerez, María del Carmen Cerón-García, Alexandra Muñoz-Bonilla, Marta Fernández-García, Francisco García-Camacho, Emilio Molina-Grima

**Affiliations:** 1Chemical Engineering Department, University of Almería, 04120 Almería, Spain; 2Research Center in Agrifood Biotechnology (CIAMBITAL), University of Almería, 04120 Almería, Spain; 3Institute of Polymer Science and Technology (ICTP-CSIC), 28006 Madrid, Spain

**Keywords:** biofouling, exopolymeric substances, surfaces adsorption

## Abstract

An improved method that allows the robust characterization of surfaces is necessary to accurately predict the biofouling formation on construction materials of photobioreactors (PBR). Exopolymeric substances (EPS), such as proteins and polysaccharides, have been demonstrated to present a similar behavior to cells in terms of surface adhesion. In this work, these EPS were used to optimize parameters, such as EPS concentration or adsorption time, to evaluate accurately the adsorption capacity of surfaces and, with it, predict the biofouling formation in contact with microalgae cultures. Once the method was optimized, the characterization of seven commercial polymeric surfaces was submitted to different abrasive particles sizes, which modified the roughness of the samples, as well as protein and polysaccharide lawns, which were prepared and carried out in order to evaluate the characteristics of these substances. The characterization consisted of the determination of surface free energy, water adhesion tension, and critical tension determined from the measurement of the contact angle, roughness, surface zeta potential, and the EPS adhesion capacity of each material. This will be useful to understand the behavior of the surface in the function of its characteristics and the interaction with the solutions of EPS, concluding that the hydrophobic and smooth surfaces present good anti-biofouling characteristics.

## 1. Introduction

Surfaces immersed in water are subjected, to a greater or lesser extension, to the adhesion and accumulation of microorganisms whose typology depends on the aqueous environment and the physico-chemical properties of surfaces. The biofilm formed is known as biofouling [1,2]. Biofouling is an undesirable phenomenon because of its negative effect on the performance of the devices or systems on which it appears [3,4]. In the case of photobioreactors (PBR), for microalgae cultivation, biofouling reduces irradiance penetration inside the PBR, decreasing biomass productivity and, therefore, the photosynthetic efficiency of the cultivation system [3]. Moreover, biofouling becomes a source of ongoing infection for invasive microorganisms, whose thorough cleaning and disinfection are impossible as nooks and crannies remain inaccessible to cleaning agents [5,6]. The above leads to premature culture stoppage and a reduction in the lifetime of the material [3].

Fouling is mainly determined by surface properties, including surface energy, roughness, and wettability [7]. There are different antifouling strategies based on polymeric materials with distinct surface wettability. The antifouling mechanism strongly relies on the environment due to the intricate interactions between foulant, antifouling materials, and solvent media [8,9].

The biofouling formation process in PBRs is similar to that reported in other surfaces immersed in water. It is generally divided into two sequential steps [1,4]: the first one consists of (i) a conditioning film formation through an initial accumulation of organic molecules physically adsorbed on the surface, and (ii) the second one implies the adhesion of microalgae on the conditioning film and the proliferation of the attached cells. The main source of film-forming compounds comes from the excretion of sticky exopolymeric substances (EPS) by cultured microalgae (e.g., proteins, polysaccharides, glycoproteins, etc.). EPS attached to the PBR walls increases the biofouling propensity of the surface [10]. 

In recent studies, the microalgal biofouling propensity of different smooth materials and coatings was satisfactorily predicted by a short-term protein test using bovine serum albumin (BSA) as a model protein [6,10]. The BSA adhesion and microalgal biofouling intensities resembled the Baier curve. This short-term BSA test was proposed as a valuable tool for selecting the materials for the construction of PBR [6,10]. BSA is a globular protein with an average size of 66 kDa, which can adopt different conformations in the function of the changes in pH or the ionic strength of the solution [11]. Its interaction and/or adsorption onto surfaces has been widely investigated due to its importance in biomedical technologies [12]. 

Polysaccharides are also abundantly excreted by microalgae and are presumably important contributors to the formation of the conditioning film [3,10]. However, the adhesion of microalgal polysaccharides on the surfaces has been barely studied. Since starch is a popular polysaccharide excreted by some types of microalgae under stress conditions [13,14], potato starch (PST) is used as a standard polysaccharide. Understanding the interaction between the surfaces and EPS requires the preparation of EPS lawns to study the characteristics of these substances in terms of surface free energy or surface charge determined from the measurement of the contact angle.

As mentioned above, the roughness of the surface can play an important role in the adhesion of organisms on the surface. Surface irregularities, known as anfractuosity, increase the surface area that can be colonized and also increase the number of places to adhere [15,16]. However, it is unknown whether cell adhesion on rough surfaces is mediated by the roughness itself, by the conditioning film, or by both. 

The objective of this study was to develop an analysis method that allows, on the one hand, the highest adsorption of EPS and, on the other hand, their highest extraction for its subsequent measurement. Therefore, for the optimization of the protein adhesion method, BSA was used as a reference protein and PST as the reference polysaccharide. Subsequently, diverse materials provided by different manufacturers with a wide range of hydrophobicity were subjected to particle abrasion treatment to increase their roughness. These materials were characterized by electronic scanning microscopy (SEM) analysis, surface zeta potential, contact angle measurement, and the adhesion capacity of EPS. The surface roughness influence on the EPS adsorption was also analysed, and it was checked whether or not they followed the biocompatibility theories of Baier [17] and Vogler [18]. This study allows the choosing of surfaces that have the most suitable properties to build efficient PBRs, where the formation of biofouling is reduced.

## 2. Results and Discussion

### 2.1. Protein and Polysaccharide Lawns Characterization

The surface properties of BSA and PST lawns obtained from the contact angle measurements are presented in Table 1. BSA can be considered hydrophilic (τ_0_ > 30 mJ m^−2^), according to the criteria of Vogler [18]. The results from the BSA lawn characterization do not vary with the support used so that in both types of material, the protein will adhere despite the differences in hydrophobicity of them. These results are in consonance with those reported by Białopiotrowicz and Jańczuk [19].

The water contact angle (*θ_w_*) measurements for the PST lawn after an immersion time of 2 h (see Table 1) differed nearly 10° between both supports. For that reason, another PST lawn was prepared, but in this case, the surfaces were immersed for 24 h in the PST solution to test if it was necessary for more time to obtain a carbohydrate lawn. After this new treatment, it can be observed that, for the PST lawn on the PMMA support, the three contact angles at 2 h and 24 h of immersion were similar, with *θ_w_* around 70° in concordance with the literature [20,21]. By contrast, the *θ_w_* value for the PST lawn on GS after 24 h of immersion (66 ± 4°), although higher than 2 h, continued to be below 70°. These results indicate that PST needs more than 24 h to form a lawn over a hydrophilic surface such as borosilicate glass, compared to a hydrophobic surface such as PMMA. This may be due to the poor PST water wettability (*τ*_0_ < 30 mJ m^−2^) [18], which implies a preference towards non-polar surfaces such as PMMA.

### 2.2. Optimization of the EPS Adhesion Method

Figure 1 displays the percentage recoveries of the BSA and PST attached, relative to the recovered maximum values of both, using the strategy described in the Materials and Methods, which aimed to improve the determination of EPS adsorbed on the surfaces. Figure 1A,B represent the effect of two factors on the relative percentage recovery of BSA and PST: (A) the effect of the BSA and PST concentration in the solution used for the immersion of the surfaces during the adsorption step (in this case, the adsorption and extraction are performed at 100 rpm); (B) the duration of the adsorption step. In both assays, the duration of the treatments to detach the BSA and PST adsorbed, i.e., the orbital shaking and ultrasonication, was set at 2 h and 30 min, respectively. Overall, the maximum percentages of both detached EPSs were observed at surfaces subjected to EPS concentrations above 1.5 g L^−1^ and an adsorption time of around 24 h. The method based on ultrasonication provided detachment levels of BSA much lower than that of orbital shaking (*p* < 0.05). Ultrasounds may have modified the functionality of BSA. It has been reported that high-intensity ultrasound causes changes in the structure of BSA in an aqueous solution, leading to surface activity, surface hydrophobicity, and surface charge increases [22]. The magnitude of the structural and functional changes depended on the treatment intensity. In general, proteins subjected to a high-intensity ultrasound significantly reduced their size and hydrodynamic volume, presumably because of the hydrodynamic shear forces associated with ultrasonic cavitations (the sonomechanical effect) and the rise in temperature at the site of the bubble collapse [23]. Cavitations give rise to transient bubbles whose collapse produces reactive free radicals, especially hydroxyl radicals, which contribute to structural changes in the proteins (sonochemical effect) [24,25]. Both sonochemical and sonomechanical effects may lead to fully deactivated proteins in an irreversible unfolded state. In the most unfavorable scenario, the increase in local temperature may cause protein denaturation. How all these changes could affect the BSA desorption and their quantification is a complex matter. As a result, it can be concluded that the method is not suitable for the quantification of proteins adsorbed on a surface.

Figure 1C shows the influence of the orbital shaking rate during the adsorption step. For the shaking rates greater than 100 rpm, the adhered protein concentration remained stable, hovering around 100%, so this was selected as the optimal orbital shaking rate. For PST adsorption, there was a maximum of 100 rpm of orbital shaking, as the percentage of PST extracted reduced until 60% when the orbital shaking rate was 150 rpm. Proteins and polysaccharides are both sensible compounds with high shear rates [26], especially polysaccharides, whose structure can be modified when the shear rate exceeds 1000 s^−1^ [27], which can explain the decrease in the extracted PST at the highest shaking rates.

Finally, the extraction orbital shaking rate and time were optimized. Both experiences are represented in Figure 1D, where the maximum of the adhered BSA protein (100%) was observed at a rate of 150 rpm, keeping this constant at 2 h and 3 h so that the shortest time was chosen as the optimum. For PST, the extraction made at 100 rpm was the highest. When it was shaken at 150 rpm, the quantity of the PST recovered was less, so there was degradation at this level of shaking. In terms of time, there was a maximum level of polysaccharide extracted at 2 h.

### 2.3. Materials Physico-Chemical Characterization

Table 2 shows the surface characterization of the materials studied and obtained from the different manufacturers. Following the Vogler criteria [18], the range of materials tested has been very varied, from poorly or moderately water-wettable materials, such as PE or PVC, PC, or PETG (*τ*_0_ < 30 mJ m^−2^), up to high water-wettable materials, such as PMMA or GS (*τ*_0_ > 30 mJ m^−2^). Furthermore, all initial materials provided by the manufacturers can be considered smooth (Ra < 0.6 µm) [28].

When comparing the physico-chemical properties of materials provided by different manufacturers, it was observed that there were no huge differences between them, even though the case of PMMA was appealed. These materials did not show a significant difference between the transparent PMMA provided by Transglass^®^ (Asturias, Spain) and Ferplast^®^ (Terrassa, Barcelona, Spain). However, there was a difference within the Ferplast^®^ manufacturer between transparent and black PMMA, the transparent being hydrophilic (*τ*_0_ = 34.1 mJ m^−2^) while the black one was very hydrophobic (*τ*_0_ = −4.7 mJ m^−2^). This may be due to the use of additives, either with the main objective of changing the material colour and, as a consequence, causing changes in its surface properties or changing some other physical or chemical properties [29].

Table 3, Table 4, Table 5, Table 6, Table 7 and Table 8 collect the surface characterization of each material with different roughness. Results for GS (Normax ^®^) are shown in Table 3, where it is possible to see that the increase in roughness did not significantly affect the in-water contact angle and, subsequently, the in-water adhesion tension. In the case of PC from Ferplast^®^ (see Table 4), it can be observed that when the surface roughness was increased, the water contact angle increased, obtaining surfaces that were more hydrophobic (decreasing *τ*_0_), even with *τ*_0_ negatives values, as for the values of Ra at 1.2 and 1.6 µm. The *γ_s_* decreased with the roughness.

For PE obtained from Ferplast^®^, its characterization with different roughnesses (see Table 5) shows that increasing the surface roughness augments the water contact angle until 0.8 µm, the value from which it starts to decrease. Similar behaviour is found for *τ*_0_ with an increment in the hydrophobic character when roughness augments. However, the change is not very pronounced for the *γ_s_* and *γ_c_* parameters.

In the case of PETG from Ferplast^®^ (see Table 6), it can be seen again that increasing the surface roughness increases the water contact angle as well until 0.8 µm. However, for this material, it can be observed that *τ*_0_, *γ_s_*, and *γ_c_* decrease with the roughness. The surface characterization for the transparent PMMA from Transglass^®^ is presented in Table 7, where it can be observed that the increasing roughness increases the water contact angle, making the *τ*_0_ decrease. The last material studied, the surface characterization of which can be seen in Table 8, is PVC from Transglass^®^. In this case, the water contact angle is almost constant, while *γ_s_* and *γ_c_* decrease with roughness up to a Ra of 0.8 µm.

Accordingly, and in general terms, the increase in material roughness increases the water contact angle and decreases the *τ*_0_, *γ_s_*, and *γ_c_*.

Figure 2 shows the SEM micrographs of the different material surfaces with roughness. It is known that, depending on the material, the scratch behaviour is different. In general, ductile plastic materials tend to have a regular appearance of the scratch channel, whereas brittle plastic materials often show an irregular scratch channel with a matrix performance debris and crack formation [30,31]. As can be seen from Figure 2, all surfaces, with the exception of PMMA, PC, and GS, exhibit more brittle damage than ductile. This behaviour could influence the adhesion behaviour as it will be further considered.

Once the methods of the adhesion of proteins and polysaccharides were optimized, it was employed to effectively determine the EPS adhesion capacity of each material and the effect when the roughness was modified from non-treated-to-treated surfaces. Figure 3 represents the average relative percentage of EPS adhered with the method of a 95% lowest significant difference (LSD) of Fisher. In the first place, in Figure 3A, the different materials studied for each roughness level are grouped, and Figure 3B shows all the roughness levels studied for each material. Characteristics of the material itself, as well as the modification of the surface roughness by applying the different abrasive particles, influences the EPS adhesion capacity.

In general, the large adsorption of the BSA protein is reached when surfaces have an average roughness of between 0.8 and 1.2 μm with values of 50% more than the control (GS, with a protein adsorption capacity of 3.0 ± 0.1 µg cm^−2^ of BSA, is taken as the control with 100% of the adsorption as is explained in the experimental section). The increment of the roughness to the upper levels (1.6 µm) does not imply more protein adsorption but rather a decrease in this capacity for all the surfaces tested. The ductile damage behaviour of some materials, such as PMMA and PC, as seen in Figure 2, could explain the reduction in adhesion for the highest roughness levels. In terms of materials (see Figure 3B), it can be observed that an increment in the BSA adsorption by PETG, which is the material whose surface free energy is the highest, while PE has the lowest surface free energy (35.7 mJ m^−2^), presents the minimum value of adhesion [10].

On the other hand, when the capacity of PST adhesion is tested, similar profiles are shown regardless of the size of the abrasive particle used (see Figure 3A). There is slightly higher adsorption of PST when the surfaces have an average roughness of 1.2 μm. Having in mind the material surface, PC, PVC, and PE can avoid adhesion (see Figure 3B). However, the PMMA surface gives place to the maximum level of PST that is adhered to. This could be due to the fact that this surface presents the most important contributor to brittle damage, which makes the surface more irregular with channels that have similar sizes of PTS; therefore, they are able to adhere to the polysaccharide.

Regarding the surface zeta potential (*ζ_s_*) (Figure 4), notable differences between the different tracer materials used can be appreciated; hence, the characterization of the materials in terms of their surface zeta potential depends on the fluid that is in contact with them. Nevertheless, all the surfaces tested present negative zeta potential values, which imply that all the surfaces are negatively charged with the independence of the tracer material used. With the use of the zeta potential standard of 50 ± 5 mV, GS and PETG are the materials with the lowest surface zeta potential reaching values lower than −50 mV [32], and the rest of the materials present similar values, which are around −40 mV, which is in consonance with values reported by other authors [33].

When BSA solutions are used as tracer materials (with a zeta potential of −9 ± 2 mV), an increase in the measured values can be observed, which continues rising with the media conductivity when a concentration of NaCl is added. This phenomenon was reported previously by Zeriouh et al. [34], where a linear relationship was determined between the *ζ* surface of the glass and the negative logarithm of ionic strength using a solution of microalgae as a tracer material with similar values of *ζ_s_* to the BSA solution used in this study. Moreover, in this case, when the influence of the salinity in the media was tested, the differences in the surface zeta potential values among the materials were diminished. That is, while the solution did not contain salt, the media surface zeta potential of all materials had a standard deviation of 5 mV, while when the solution was prepared with a concentration of 20 mM of NaCl, the standard deviation was reduced up to 2 mV.

Therefore, the effect of salinity shields the influence of the material on the dissolved particles in the medium. When the ionic force is bigger, the ζ_s_ tends to assume values close to −10 mV, as has been previously described [34].

The biocompatibility theories of Baier [17] and Vogler [18] can have interesting results in the study of the behaviour of biofouling formation on surfaces with different characteristics [10,35]. Figure 5 shows the relative adhesion of BSA (Figure 5A) and PST (Figure 5B), both as a function of the water adhesion tension (Vogler’s theory) and at the critical surface tension (Baier’s theory) for the different materials tested when subjected to different roughness levels. As it was mentioned previously, the characteristics in terms of the wettability of the surfaces were modified with their roughness, and fewer differences were observed among the materials with water adhesion tension (and subsequently with critical surface tension) while the roughness of the material increased. In Figure 5A, a minimum BSA adhesion can be observed with the non-treated surface (0.1 µm), which is between the minimum area described by Baier. When the roughness is incremented, the differences among the water adhesion tension for the materials are shorter, but the minimum BSA adhesion observed for the lower values of critical adhesion tension coincide, in some cases, with Baier’s minimum.

For the adhesion of PST in terms of biocompatibility theories (see Figure 5B), with the non-treated surfaces, it is observed that minimum values of adhesion for the lowest critical surface tension coincide with Baier’s minimum. This tendency is followed by the materials treated with abrasive particles with higher roughness.

Surfaces could acquire amphiphilic properties after their exposure to seawater, reaching values of adhesion closer to Vogler’s minimum (*τ*_0_ of around 35 mJ m^−2^) [10,35]. As it is mentioned above, the media diminish the differences between the surface materials (see Figure 4). Therefore, it would be an interesting time ahead to study the adhesion behaviour of surfaces in contact with seawater cultures.

## 3. Materials and Methods

### 3.1. Protein and Polysaccharide Lawn Preparation

To elaborate on protein and polysaccharide lawns, the method proposed by Białopiotrowicz and Jańczuk [19] was followed. The protein and polysaccharide lawns were prepared with bovine serum albumin (BSA) (Sigma Aldrich) and potato starch (PST) (Panreac), respectively. The BSA and PST concentrations in the solution were 15 g L^−1^. This concentration is well above the 2 g L^−1^ threshold value observed for BSA [19]: values below 2 g L^−1^ may lead to a drop in the contact angle values, as this concentration is also needed for the monolayer coverage of a PMMA surface [19].

Lawns were prepared on the supports of poly(methyl methacrylate) (PMMA) and borosilicate glass (GS): both materials widely used in the PBR’s manufacture for culturing microalgae. After the adsorption of BSA and PST on the supports during a specific time that warranted the complete adsorption of the substances, these were kept dry for at least 48 h at room temperature before measurements. The Lifshitz-van der Waals components and the values of the electron-acceptor and electron-donor parameters of the acid-base components of the films were calculated from the obtained contact angles following the calculation methodology proposed by Zeriouh et al. [28].

### 3.2. Surfaces Preparation

The rigid materials used in this study were polycarbonate (PC), polyethylene (PE), poly(ethylene terephthalate glycol-modified) (PETG), poly(methyl methacrylate) (PMMA), poly(vinyl chloride) (PVC), and borosilicate glass (GS). The GS used was obtained from Normax^®^ (Marinha Grande, Portugal), PE and PETG were provided by Ferplast^®^ (Terrassa, Barcelona, Spain), and PC, PMMA, and PVC were from Transglass^®^ (Asturias, Spain) and Ferplast^®^ (Terrassa, Barcelona, Spain). All the materials were transparent except for PE and PVC from Ferplast^®^, which were opaque, and all of them were cut in rectangular coupons (2.5 cm × 7.5 cm × 0.3 cm, except glass which had a thickness of 0.1 cm).

In order to obtain a different roughness of each material to test the influence of this parameter on the exopolymeric substances (EPS) adhesion, proteins and polysaccharides surfaces were ground using abrasive particles that were sized between 30 and 260 µm with a gradation of P-500, P-220, P-100, and P-60. The combination of the different abrasive size particles allowed an average surface roughness to be achieved (R_a_: the arithmetic media deviation from the mean line within the assessment length) of 0.4, 0.8, 1.2, and 1.6 µm. The control surfaces of each material were not subjected to abrasive treatment, and they could be considered smooth (values of Ra below 0.6 µm) with roughness values ranging between 0.03 and 0.10 µm. Roughness was determined with a rough meter (PCE-RT 11, PCE Ibérica S.L., Albacete, Spain) with a 1 mm scan length and a 0.111 µm/sample resolution and after the preparation of surfaces, the scanning electron microscopy images of the different material surfaces were performed using a Philips XL30 Scanning Electron Microscope with an acceleration voltage of 25 kV. The samples were coated with 5 nm of gold/palladium (80/20) prior visualization.

After the abrasion, the surfaces were cleaned following the protocol described by Ozkan et al. [36] in the case of GS; PMMA, PE, and PETG were cleaned by the method used by Ruiz-Cabello [37]. The rest of the materials were washed with Alconox^®^ and abundant deionized (DI)and sterilized water (Elix 35 ^®^, Merck Millipore, Madrid, Spain) with a resistivity of 5 MΩ at 26 °C, avoiding the use of hexane, which can cause damage to the surfaces [35].

### 3.3. Surfaces and Lawns Physico-Chemical Properties

Contact angles on the different materials and lawns, as described previously, were measured by the sessile drop technique using a goniometer (Drop Shape Analyzer DSA25). For surface characterization, the solvents used were water and formamide as polar liquids and diiodomethane as an apolar liquid. From the contact angles between the surfaces and these solvents, the surface energy components (*γ_s_^LW^*, *γ_s_^+^*, *γ_s_^−^*, *γ_s_^AB^*), the change in free energy of cohesion (Δ*G_coh_*), the water adhesion tension (*τ*_0_), and the critical surface tension (*γ_c_*) were calculated as described earlier [28].

The surfaces were subjected to their surface zeta potential measurement using a dip cell device, ZEN100 (Malvern, UK), for a Zetasizer Nano ZS device (Malvern, UK) equipped with a He–Ne laser (633 nm) as the light source. The surface zeta potential measure indicates the stability of the interaction of the suspension and the material estimated from the electrophoretic mobility. Rectangular pieces of the material samples of 4 × 5 mm and no more than 1.5 mm thick were attached to the sample holder using Araldite^®^, Cey’s, Spain, and held between two palladium electrodes. The cell was then submerged into a conventional glass cuvette containing a zeta potential transfer standard of −50 ± 5 mV provided by Malvern Instruments Ltd. (UK) and into a protein solution of BSA 0.5%*w*/*w*, which is well-defined, including the molecular composition and physical conditions [19] with three concentrations of salt NaCl from 0 mM until 20 mM to check the influence of the solution conductivity on the measurement of the surface zeta potential.

The electro-osmotic flow at the sample surface usually decreases with an increasing distance from the surface, where the tracer mobility is dominated by the electro-osmotic surface flow at short distances from the surface and by the electrophoretic motion of the tracer itself at large distances. Thereby, measurements were made at distances ranging from 125 μm to 625 μm and a further measure at 1000 μm where the electro-osmotic does not have any influence. By plotting the reported zeta potential as a function of displacement from the surface, it is possible to calculate the zeta potential at the surface (*ζ_s_*) by extrapolating the graph to zero and applying Equation (1).
(1)$$\zeta_{s}=-intercept+\zeta_{m}$$
where *ζ_m_* is the tracer zeta potential recorded far from the surface, at 1000 μm.

### 3.4. Optimization of Rapid Protein Adhesion Method

The optimization of the protein adhesion method was focused on the adsorption and extraction steps of protein. The reference surface used was borosilicate glass (Normax^®^) provided by the manufacturer in the form of rectangular coupons (2.5 cm × 7.5 cm × 0.1 cm) with an area of 18.75 cm^−2^. The coupons were cleaned following the protocol described by Ozkan et al. [36]. Then, they were immersed in an aqueous BSA solution prepared in PBS at a given concentration (pH 7.2–7.4) and kept at 25 °C during the assay. BSA was used as a model protein [38]. Transparent 45 mL polypropylene Petri dishes, with an 85 mm internal diameter (*d*) and a 9 mm height (*h*) of the solution at rest, were used as incubation vessels. Petri dishes were placed on an orbital shaker with a 3 cm orbital diameter (*r*) (OVAN Maxi, OL30-ME). The shaker produces a circular movement in the horizontal plane, thus inducing a wave of culture medium that circles the well. The peak of the wave rotates at the angular velocity of the shaker platform. The resultant movement of the fluid exposes the coupons, located on the base of the Petri dishes, to shear stress. Non-breaking waves with full coverage of the bottom of the plates were ensured. The following operational variables were studied: (i) the concentration of the BSA solution (0.5, 1, 1.5, and 2 g L^−1^); (ii) immersion time of absorption (4, 8, 12, 21, 24, 36, and 48 h); (iii) orbital shaking rate (50, 100 and 150 rpm). The average shear rate at the bottom of an orbiting Petri dish was estimated from Equation (2) [39]:(2)$$\gamma_{\omega }=\frac{d}{2\mu }\cdot \sqrt{\rho \mu \cdot {\left(2\cdot \pi \cdot f\right)}^{3}}$$
where *ρ* is the fluid density (1000 kg m^−3^, *μ* is the fluid viscosity (10^−3^ kg m^−1^ s^−1^), and *f* is the frequency of orbit. The assayed orbital shaking rate (50, 100, and 150 rpm) corresponded to *γ_ω_* values of 329 s^−1^, 931 s^−1^, and 1712 s^−1^, respectively.

Once the adsorption step was completed, the coupons were rinsed with fresh PBS and transferred to clean Petri plates, where they were washed in an aqueous solution of 1% *w*/*w* sodium dodecyl sulphate (SDS) to detach the adsorbed BSA. The following operational variables were studied: (i) the procedure of BSA detachment (ultrasound-assisted or shaking-assisted); (ii) treatment time (5, 10, 15, and 20 min for ultrasound; 1, 2, and 3 h for shaking); (iii) the shaking rate for a shaking-assisted detachment (50, 100, and 150 rpm). The ultrasonic-assisted detachment was carried out in an ultrasonic cleaning bath (JP Selecta™ 3001208, 110 W, JP Selecta S.A, Spain).

The amount of BSA adhered to on each case was quantified by a Bicinchoninic acid (BCA) protein assay kit (Novagen, Merck) at 562 nm, as described elsewhere [38].

### 3.5. Optimization of Rapid Polysaccharide Adhesion Method

Analogously to the method described above for protein adhesion, a method was set up to measure the polysaccharide adsorption on surfaces. PST was used as the polysaccharide standard. GS (Normax^®^) was used as the support with dimensions of 7.5 cm × 2.5 cm × 0.1 cm. For the adsorption step: (i) four different concentrations of PST dissolved in phosphate buffer (PBS) from 0.5 to 2 g L^−1^ were studied. Then, (ii) eight adsorption times (from 4 to 60 h) and (iii) three orbital shaking rates (50, 100, and 150 rpm) were tested. On the other hand, the extraction step was carried out by putting into contact the surface with the adhered PST and the PBS that must extract hydrosoluble carbohydrates. (i) Three rates of 50, 100, and 150 rpm (ii) and three different times of extraction (1, 2, and 3 h) and were tested following the method described by [40].

Once the EPS adhesion (adsorption and desorption) methods were optimized using glass as the control surface, for the rest of the materials, their capacity of EPS adhesion was determined, expressing it as a function that corresponded with the control following Equation (3).
(3)$$\%\;adhered\;EPS=\frac{\frac{m_{EPS\;adhered\;on\;surface}}{A_{surface}}}{\frac{m_{EPS\;adhered\;on\;GS}}{A_{GS}}}$$

### 3.6. Statistical Analysis

The analysis of variance (ANOVA) was performed to detect any significant differences between the factors at *p* < 0.05. The software used was Statgraphics Centurion 18 (StatPoint, Herndon, VA, USA).

## 4. Conclusions

The surface EPS adhesion capacity method was optimized with a maximum adhesion for a solution concentration of 1.5 g L^−1^, in contact with the surface for 24 h at 100 rpm for both protein and polysaccharide and 2 h for the extraction by shaking at 150 rpm in the case of BSA and 100 rpm for PST. Optimized methods led to the characterization of the surfaces, observing a relationship between the total surface free energy and protein or polysaccharide adhesion capacity. The increase in the roughness using abrasive particles maximises the capacity of adhesion in the protein, but a maximum is reached with a roughness of around 0.8–1.2 µm. In the case of polysaccharides, a major roughness does not cause an increment in the adhesion capacity of the materials. BSA, a hydrophilic protein, can adhere to any surface in 2 h no matter what its hydrophobicity, while PST, which is a hydrophobic carbohydrate, needs more time to form a lawn over a polar surface (as GS) than over an apolar surface (as PMMA). In addition, the surface charge is affected by the tracer material and the salinity of the media. Therefore, in forthcoming studies, the influence of saline media on EPS adhesion will be investigated to understand the behaviour of the biofouling character of materials in seawater-simulated media. In general, it is shown that hydrophobic materials whose roughness does not exceed 0.6 µm, as is the case with these smooth and transparent PE or PVC, could be good candidates for PBRs construction. Nevertheless, other characteristics, such as optical and mechanical properties, must be taken into consideration.

## Figures and Tables

**Figure 1 ijms-23-13924-f001:**
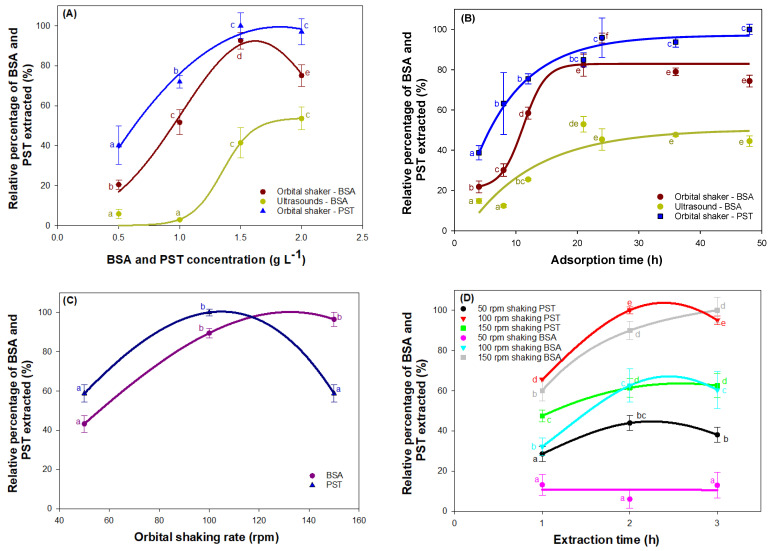
Relative percentage of protein and starch extracted by shaking and ultrasound (**A**) at different concentrations of added protein and polysaccharide (adsorption at 100 rpm and extraction at 100 rpm); (**B**) at different adsorption times; (**C**) at different orbital shaker speeds during adsorption and (**D**) at different times and orbital speeds of extraction by shaking and ultrasound. The relative percentage is calculated by taking as reference the highest concentration of BSA measured in each test. Data points are averages, and vertical bars are the standard deviation (SD) for triplicate samples. The lowercase letters represent significant differences, with a *p* < 0.05.

**Figure 2 ijms-23-13924-f002:**
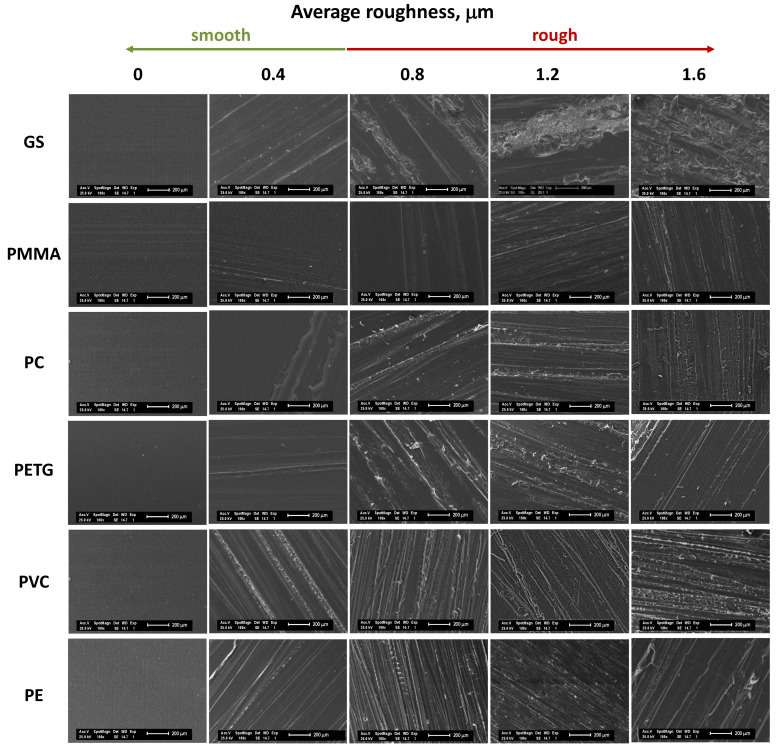
SEM images of the different material surfaces and roughness.

**Figure 3 ijms-23-13924-f003:**
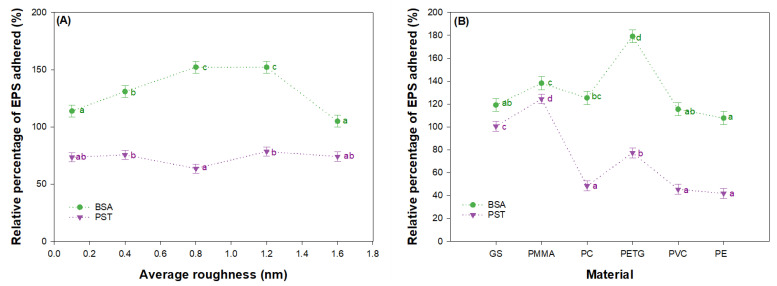
Relative percentage of BSA and PST adhered (**A**) as a function of the roughness level from 0.1 µm to 1.6 µm grouping all the materials tested and (**B**) by glass (GS), poly(vinyl chloride) (PVC), poly(ethylene terephthalate glycol) (PETG), polycarbonate (PC), poly(methyl methacrylate) (PMMA) and polyethylene (PE), grouping roughness levels from 0.1 µm to 1.6 µm. Data points are averages, and vertical bars are standard deviation (SD) for triplicate samples. The lowercase letters represent significant differences, with a *p* < 0.05.

**Figure 4 ijms-23-13924-f004:**
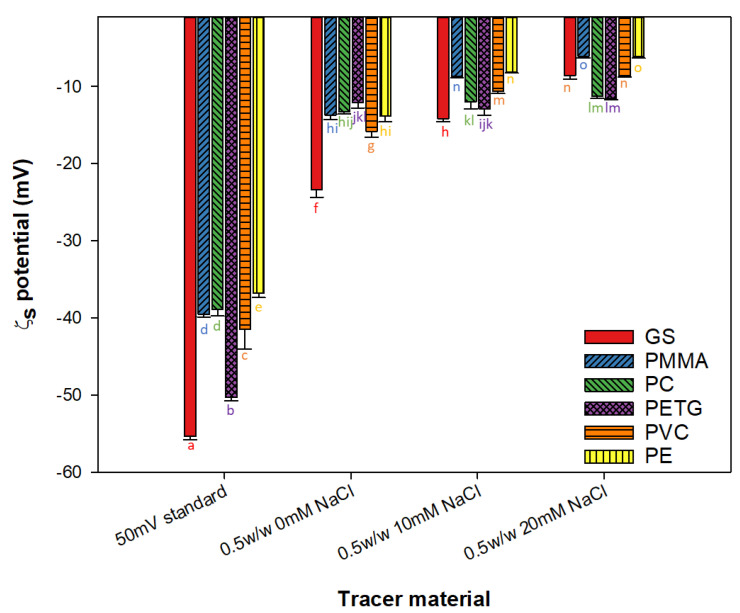
Surface zeta potential of glass (GS), polyvinylchloride (PVC), poly(ethylene terephthalate glycol) (PETG), polycarbonate (PC), poly(methyl methacrylate) (PMMA), and polyethylene (PE), using as a tracer material a 50 ± 5 mV standard, and a solution of 0.5% *w*/*w* BSA with 0, 10, and 20 mM NaCl concentration. Data points are averages, and vertical bars are standard deviation (SD) for triplicate samples. The lowercase letters represent significant differences, with a *p* < 0.05.

**Figure 5 ijms-23-13924-f005:**
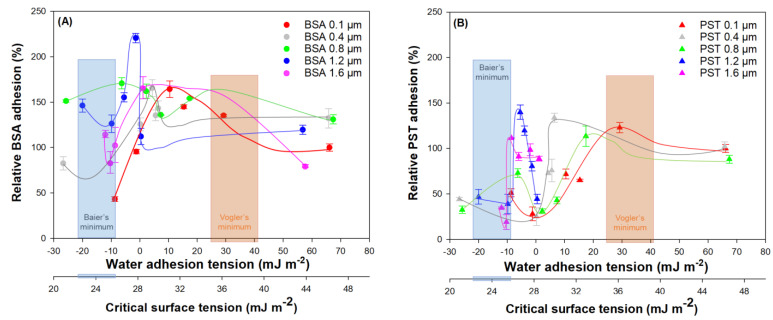
Relative adhesion on different polymeric surfaces with different roughness of (**A**) BSA and (**B**) PST as a function of water adhesion tension and critical surface tension according to Baier and Vogler biocompatibility theories.

**Table 1 ijms-23-13924-t001:** Effect of the support and immersion time on the contact angles, cohesion energy, free energy, water adhesion tension, and critical surface energy of the EPS lawns prepared.

Lawn	Support	Immersion Time (h)	Contact Angles (°)	Interaction Components and Surface Properties (mJ m^−2^)
*θ_w_*	*θ_F_*	*θ_D_*	∆Gcoh	γs	*τ* _0_	γc
BSA	PMMA	2	18 ± 4	25 ± 3	39 ± 2	36.9	50.7	69.1	47.2
BSA	GS	2	20 ± 3	30 ± 4	39 ± 2	42.2	47.5	68.6	47.1
PST	PMMA	2	71 ± 4	58 ± 3	31 ± 4	−26.7	47.6	24.3	43.7
PST	GS	2	59 ± 3	54 ± 2	43 ± 5	−0.6	38.4	37.4	38.3
PST	PMMA	24	72 ± 2	55 ± 3	33 ± 3	−36.7	43.5	23.0	42.8
PST	GS	24	66 ± 4	54 ± 4	43 ± 3	−20.5	39.9	29.6	29.6

(*θ_w_*) contact angle between the surface and the polar solvent 1 (water); (*θ_F_*) contact angle between the surface and polar solvent 2 (formamide); (*θ_D_*) contact angle between the surface and the apolar solvent (diiodomethane); (∆Gcoh) the free energy variation; (*τ*_0_) the water adhesion tension; (γs)  the surface free energy; and (γc) critical surface tension. Results are presented as the average ± SD (*n* = 3). A surface is considered smooth for Ra values below 0.6 μm.

**Table 2 ijms-23-13924-t002:** Results for the contact angles, surface interaction force components, cohesion energy, free energy, water adhesion tension, and critical surface energy for different materials.

.	Ra (µm)	Contact Angles (°)	Interaction Components and Surface Properties (mJ m^−2^)
*θ_w_*	*θ_F_*	*θ_D_*	∆Gcoh	γs	*τ* _0_	γc
PC Transglass^®^	0.04 ± 0.01	84 ± 1	69 ± 1	28 ± 1	−42.2	51.3	8.2	45.0
PC Ferplast^®^	0.05 ± 0.01	78 ± 1	70 ± 1	26 ± 1	−24.7	57.1	15.3	45.6
PE Ferplast^®^	0.10 ± 0.06	97 ± 1	84 ± 1	55 ± 1	−49.3	35.7	−8.7	31.5
PETG Ferplast^®^	0.03 ± 0.01	81 ± 1	79 ± 1	27 ± 1	−17.2	64.4	10.4	45.4
Transparent PMMA Transglass^®^	0.03 ± 0.01	66 ± 1	61 ± 2	43 ± 3	−8.3	42.0	29.2	37.8
Black PMMA Ferplast^®^	0.02 ± 0.01	94 ± 1	76 ± 1	32 ± 1	−54.6	48.4	−4.7	43.5
Transparent PMMA Ferplast^®^	0.03 ± 0.01	62 ± 1	58 ± 2	31 ± 2	−4.0	52.4	34.1	37.4
Grey PVC Ferplast^®^	0.02 ± 0.01	94 ± 1	79 ± 1	49 ± 2	−49.7	39.3	−4.5	35.1
Transparent PVC Transglass^®^	0.03 ± 0.01	91 ± 1	64 ± 4	30 ± 2	−80.2	45.0	−1.2	44.3
GS Normax^®^	0.06 ± 0.01	25 ± 1	26 ± 3	38 ± 3	30.4	50.9	66.1	56.7

(*θ****_w_***) contact angle between the surface and the polar solvent (water); (*θ_F_*) contact angle between the surface and polar solvent (formamide); (*θ_D_*) contact angle between the surface and the non-polar solvent (diiodomethane); (∆Gcoh) the free energy variation; (*τ*_0_) the water adhesion tension; (γs)  the surface free energy; and (γc) critical surface tension. Results are presented as the average ± SD (n = 3). A surface is considered smooth for Ra values below 0.6 μm.

**Table 3 ijms-23-13924-t003:** Results of the contact angles, surface interaction force components, cohesion energy, free energy, water adhesion tension, and critical surface energy at different roughness for GS surface.

Ra (µm)	Contact Angles (°)			Interaction Components and Surface Properties (mJ m^−2^)			
	*θ_w_*	*θ_F_*	*θ_D_*	$∆G_{coh}$	$\gamma_{s}$	*τ* _0_	$\gamma_{c}$
0.06 ± 0.01	25 ± 1	26 ± 3	38 ± 3	30.4	50.9	66.1	56.7
0.40 ± 0.04	25 ± 3	18 ± 2	44 ± 1	21.8	54.9	65.8	56.5
0.80 ± 0.02	22 ± 1	18 ± 1	43 ± 2	25.4	54.7	67.4	57.2
1.20 ± 0.05	24 ± 3	15 ± 1	49 ± 4	20.5	56.2	66.4	56.8
1.60 ± 0.06	20 ± 2	14 ± 2	47 ± 3	24.8	56.2	68.6	57.7

**Table 4 ijms-23-13924-t004:** Results of the contact angles, surface interaction force components, cohesion energy, free energy, water adhesion tension, and critical surface energy at different roughness for PC surface.

Ra (µm)	Contact Angles (°)			Interaction Components and Surface Properties (mJ m^−2^)			
	*θ_w_*	*θ_F_*	*θ_D_*	$∆G_{coh}$	$\gamma_{s}$	*τ* _0_	$\gamma_{c}$
0.05 ± 0.01	78 ± 1	70 ± 1	26 ± 1	−24.7	57.1	15.3	45.6
0.40 ± 0.14	86 ± 3	66 ± 1	49 ± 1	−59.5	35.2	5.5	34.6
0.80 ± 0.17	84 ± 3	67 ± 2	41 ± 1	−51.2	41.6	7.3	39.4
1.20 ± 0.25	98 ± 2	72 ± 5	48 ± 5	−87.3	35.4	−9.8	35.3
1.60 ± 0.12	98 ± 3	70 ± 2	53 ± 5	−86.7	33.1	−10.3	32.7

**Table 5 ijms-23-13924-t005:** Results of the contact angles, surface interaction force components, cohesion energy, free energy, water adhesion tension, and critical surface energy at different roughness for PE surface.

Ra (µm)	Contact Angles (°)			Interaction Components and Surface Properties (mJ m^−2^)			
	*θ_w_*	*θ_F_*	*θ_D_*	$∆G_{coh}$	$\gamma_{s}$	*τ* _0_	$\gamma_{c}$
0.10 ± 0.06	97 ± 1	84 ± 1	55 ± 1	−49.3	35.7	−8.7	31.5
0.40 ± 0.18	112 ± 4	87 ± 3	51 ± 2	−82.4	34.1	−26.7	33.8
0.80 ± 0.22	111 ± 1	82 ± 3	39 ± 3	−83.6	40.0	−25.7	40.0
1.20 ± 0.05	106 ± 2	78 ± 2	42 ± 2	−92.3	39.0	−20.0	38.9
1.60 ± 0.14	100 ± 3	76 ± 1	42 ± 2	−78.1	40.0	−12.0	38.8

**Table 6 ijms-23-13924-t006:** Results of the contact angles, surface interaction force components, cohesion energy, free energy, water adhesion tension, and critical surface energy at different roughness for PETG surface.

Ra (µm)	Contact Angles (°)			Interaction Components and Surface Properties (mJ m^−2^)			
	*θ_w_*	*θ_F_*	*θ_D_*	$∆G_{coh}$	$\gamma_{s}$	*τ* _0_	$\gamma_{c}$
0.03 ± 0.01	81.3 ± 0.5	78.9 ± 0.9	27.1 ± 1.3	−17.2	64.4	10.4	45.4
0.40 ± 0.18	86.6 ± 2.2	61.6 ± 1.2	40.6 ± 1.3	−71.1	39.9	4.3	39.3
0.80 ± 0.08	95.0 ± 1.9	74.1 ± 2.3	48.0 ± 4.1	−70.2	36.5	−6.3	35.4
1.20 ± 0.17	91.1 ± 5.5	70.9 ± 3.1	64.0 ± 5.3	−59.6	28.4	−1.4	26.3
1.60 ± 0.26	89.1 ± 2.4	71.6 ± 4.5	55.5 ± 5.1	−60.1	31.2	1.1	31.2

**Table 7 ijms-23-13924-t007:** Results of the contact angles, surface interaction force components, cohesion energy, free energy, water adhesion tension, and critical surface energy at different roughness for PMMA surface.

Ra (µm)	Contact Angles (°)			Interaction Components and Surface Properties (mJ m^−2^)			
	*θ_w_*	*θ_F_*	*θ_D_*	$∆G_{coh}$	$\gamma_{s}$	*τ* _0_	$\gamma_{c}$
0.10 ± 0.03	66 ± 1	61 ± 1	43 ± 3	−8.3	42.0	29.2	37.8
0.40 ± 0.15	85 ± 4	69 ± 3	43 ± 1	−49.1	40.6	6.4	38.1
0.80 ± 0.05	76 ± 2	67 ± 1	30 ± 1	−26.5	52.9	17.4	44.4
1.20 ± 0.16	94 ± 5	66 ± 2	50 ± 3	−80.6	34.8	−5.5	34.1
1.60 ± 0.14	97 ± 4	60 ± 2	42 ± 5	−94.4	38.6	−8.6	38.6

**Table 8 ijms-23-13924-t008:** Results of the contact angles, surface interaction force components, cohesion energy, free energy, water adhesion tension, and critical surface energy at different roughness for PVC surface.

Ra (µm)	Contact Angles (°)			Interaction Components and Surface Properties (mJ m^−2^)			
	*θ_w_*	*θ_F_*	*θ_D_*	$∆G_{coh}$	$\gamma_{s}$	*τ* _0_	$\gamma_{c}$
0.03 ± 0.01	91 ± 1	64 ± 4	30 ± 2	−80.2	45.0	−1.2	44.3
0.40 ± 0.07	90 ± 3	65 ± 4	42 ± 3	−77.2	38.8	0.2	38.6
0.80 ± 0.05	88 ± 5	66 ± 5	57 ± 4	−61.0	32.7	2.2	30.6
1.20 ± 0.13	90 ± 5	71 ± 5	55 ± 5	−62.9	31.8	0.4	31.7
1.60 ± 0.18	92 ± 3	72 ± 4	46 ± 5	−64.1	37.9	−2.0	36.5

## Data Availability

Not applicable.

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
