# Peer review of "Adsorption Analysis of Exopolymeric Substances as a Tool for the Materials Selection of Photobioreactors Manufacture"

_ijms, 2022, doi:10.3390/ijms232213924_

Round 1

Reviewer 1 Report

The manuscript is very interesting, and the authors presented good experimentation and reliable results.

I am sure that this study will be useful for researchers working on this subject, however, the paper needs correction/revision.

Please also consider the following points:

·      Please re-write the abstract. Either some information is missing or given the situation is in long sentences which makes it not to understand...

·      Line 54-55: Delete the sentence. “This short term…”.

·      I was expecting the “Materials and methods” subtitle after the “Introduction” but I see that it is given later.

·      Figure 1: Modify the figure. Delete the second y-axis, and included it in the first y-axis as “BSA and PST”.

·      Figure 1: What does this a, b, or c mean? There is no explanation in the text.

·      Provide the information for deionized water (DI) water equipment such as company, country, model, resistivity, etc.

·      In addition to them, the manuscript contains some grammatical problems and must be carefully edited again in terms of grammar and writing.

·      Please see the attached file where you can find my corrections/comments, and respond to them carefully one by one.

Overall, the paper can be accepted after each comment is responded to carefully. Then, it can be considered to be published in the journal of “International Journal of Molecular Sciences”. 

Author Response

Dear Reviewer:

We appreciate the reviewer's comment. We completely agree with the reviewer, and we have changed the abstract accordingly. It has been addressed in the revised version.

All figures have been changed as suggested.

In the Figure 1 is already explained in the legend the meaning of lowercase letters as “The lowercase letters represent significant differences, with a p <0.05”.

The information of DI has been included in materials and methods.

We have corrected the grammatical problems along the manuscript. We are very grateful for reviewer corrections; indeed, they have improved the manuscript.

Reviewer 2 Report

This draft can be published as is. However, it would be better more explanation about the usage field in real life. 

Author Response

Dear Reviewer:

It has been addressed in the revised version. In introduction section has been included some phrases and, in the conclusions, too. These sections have been changed accordingly.

Reviewer 3 Report

After reviewing the manuscript, which is generally devoted to the biofouling of construction materials of photobioreactors for microalgae cultivation, I can draw the following conclusions. The article is devoted to an important problem and its scientific and scientific-practical significance is beyond doubt. The introduction, in my opinion, quite fully reflects the essence of the problem and contains links to the main modern research on this topic. Materials and methods, as well as mathematical approaches to data processing, used by the authors also do not cause comments and doubts about the reliability of the results obtained.

Interesting data were obtained from Protein and polysaccharide lawns characterization and also, in my opinion, an important conclusion the authors received was the adaptation of methods for the quantification of proteins adsorbed on a surface. It seems important and interesting to systematically compare a series of obtained and calculated data on adhesion (contact angles, surface interaction force components, cohesion energy, free energy, water adhesion tension and critical surface energy) of polysaccharides for different materials. On this basis, I believe that this material can be published in this form without significant changes. As a wish, it is worth adding that in conclusion I would like to see more clear practical thoughts, for example: such a material is good and we recommend it for use in photobioreactors.

Author Response

Dear Reviewer:

We agree with the reviewer. It has been addressed in the revised version. In conclusion section one phrase to recommend, which material can be used in photobioreactors has been included accordingly.
